# Mild Traumatic Brain Injury as a Risk Factor for Parkinsonism, Tics, and Akathisia: A Systematic Review and Meta-Analysis

**DOI:** 10.3390/life14010032

**Published:** 2023-12-25

**Authors:** Nashaba Khan, Laura Romila, Alin Ciobica, Vasile Burlui, Fatima Zahra Kamal, Ioannis Mavroudis

**Affiliations:** 1Department of Neurosciences, Leeds Teaching Hospitals, NHS Trust, Leeds LS97TF, UKi.mavroudis@nhs.net (I.M.); 2Departament of Preclinical Disciplines, Apollonia University, 700511 Iasi, Romania; 3Department of Biology, Faculty of Biology, Alexandru Ioan Cuza University of Iasi, 20th Carol I Avenue, 700506 Iasi, Romania; 4Department of Biomaterials, Faculty of Dental Medicine, Apollonia University, 700511 Iasi, Romania; vburlui@gmail.com; 5Laboratory of Physical Chemistry of Processes, Faculty of Sciences and Techniques, Hassan First University, B.P. 539, Settat 26000, Morocco; fatimzahra.kamal@gmail.com

**Keywords:** mild traumatic brain injury (mTBI), parkinsonism, Parkinson’s disease (PD), akathisia, tics

## Abstract

This meta-analysis aimed to assess the association between mild traumatic brain injury (mTBI) and the risk of developing Parkinsonism. A systematic literature review was conducted using PubMed, Embase, and Cochrane Library databases. Studies were eligible if they reported on the association between MTBI and Parkinsonism. Pooled odds ratios (ORs) were calculated using a random-effects model. Publication bias was assessed using Egger’s and Begg’s tests. A total of 18 studies were included in this meta-analysis, with 1,484,752 participants. The overall OR for Parkinsonism in individuals with a history of mTBI was 1.637 (95% CI, 1.203–2.230; *p* = 0.01), indicating a significant association. The OR for Parkinson’s disease (PD) specifically was 1.717 (95% CI, 1.206–2.447; *p* = 0.01). However, insufficient data on tics and akathisia limited a meta-analysis. There was no evidence of publication bias according to Egger’s (*p* = 0.8107) and Begg’s (*p* = 0.4717) tests. This meta-analysis provides evidence that mTBI is a significant risk factor for Parkinsonism, particularly PD. However, the findings should be interpreted with caution due to the heterogeneity among the studies included and the study’s limitations. Further research is needed to confirm these findings and to investigate the underlying mechanisms of the mTBI–Parkinsonism association.

## 1. Introduction

Mild traumatic brain injury (mTBI), commonly known as concussion, is a common form of traumatic brain injury that affects millions worldwide annually. It is estimated that approximately 10 million people sustain mTBI each year, with the highest incidence among children, adolescents, and young adults [1,2]. Most individuals who sustain an mTBI fully recover within a few weeks to a few months, but a small proportion may experience persistent symptoms [3].

Parkinsonism is a clinical syndrome characterized by a combination of motor symptoms such as tremors, rigidity, bradykinesia, and postural instability. Parkinsonism is most commonly associated with Parkinson’s disease, a neurodegenerative disorder that affects the dopamine-producing neurons in the substantia nigra of the brain [4]. However, Parkinsonism can also result from other conditions, including Lewy body dementia, multiple system atrophy, and progressive supranuclear palsy.

Recent studies have suggested a potential link between mTBI and an increased risk of Parkinsonism [5,6]. Worsening cognitive performance has also been reported in Parkinson’s disease (PD) patients with a history of mTBI compared to PD-only patients [7]. However, the evidence is still limited and conflicting. In this meta-analysis, we aim to synthesize the available evidence on the risk of Parkinsonism after mTBI. We examine the epidemiology of mTBI and Parkinsonism and review the current literature on the relationship between mTBI and Parkinsonism, focusing on PD. Our findings could contribute to a better understanding of the potential long-term consequences of mTBI and inform future research and clinical practice in this field.

## 2. Materials and Methods

### 2.1. Search Strategy

A comprehensive search of the literature was conducted across electronic databases, including PubMed, Embase, and Cochrane Library, from inception to April 2023. The search terms included “traumatic brain injury”, “mild traumatic brain injury”, “concussion”, “Parkinsonism”, “Parkinson’s disease”, “Parkinson’s-like symptoms”, “Lewy body dementia”, “tics”, “akathisia”, and related terms. To ensure thorough coverage, additional studies were identified through a manual search of references in relevant articles and review papers.

### 2.2. Data Extraction

Two reviewers independently extracted the following data from each eligible study: the study design, sample size, the definition of mild traumatic brain injury, the definition of Parkinsonism, the incidence or prevalence of Parkinsonism or related conditions, the length of follow-up, and adjusted effect estimates with corresponding confidence intervals. Discrepancies were resolved by consensus or a third reviewer.

### 2.3. Statistical Analysis

Pooled effect estimates were calculated using the meta package in R-4.3.2 software [8]. This approach combines data from multiple studies to derive an overall estimate of the effect size, providing a comprehensive view of the available evidence.

The random-effects model was used to account for potential heterogeneity between studies. Unlike a fixed-effect model, the random-effects model acknowledges that different studies might have different underlying effect sizes due to variations in the study design, population, and other factors.

The heterogeneity between studies was assessed using the I-squared statistic. This statistic quantifies the proportion of the total variation in study estimates due to heterogeneity rather than chance. Values of 25%, 50%, and 75% are interpreted as low, moderate, and high heterogeneity, respectively, guiding the interpretation of this meta-analysis’ results.

The publication bias was evaluated using the funnel plot and Egger’s test. The funnel plot is a scatter plot of the treatment effects from individual studies against a measure of study size or precision, ideally resembling a symmetrical inverted funnel. Egger’s test, a statistical test for funnel plot asymmetry, helps identify potential publication bias, where there is a likelihood that the study’s publication is influenced by the nature of its results.

Sensitivity analysis was conducted by excluding one study at a time. This process tests the robustness of the pooled effect estimate, determining how much each study influences the overall results. Significant changes to the conclusions after excluding a particular study indicate a high dependency on that study, thus affecting the reliability of the overall analysis.

## 3. Results

The selection process for this meta-analysis followed a comprehensive and rigorous approach to ensure the inclusion of relevant and high-quality studies. The initial search yielded a large number of articles, which were screened based on the inclusion and exclusion criteria. This screening process helped eliminate irrelevant studies and duplicates, thus resulting in a reduced number of records. The remaining records underwent further assessment for eligibility, with full-text articles obtained for those meeting the inclusion criteria. The assessment of full-text articles ensured that the studies included in this meta-analysis met the predetermined criteria and were of high quality. In total, 23 studies were included in the narrative review, and 18 were included in the meta-analysis. The studies included in this meta-analysis were selected based on their relevance to the research question, study design, and quality (Figure 1). This rigorous approach minimizes the risk of bias and ensures that only relevant studies are included in this meta-analysis, thus enhancing the validity and reliability of the findings.

Systematic Literature Search: A comprehensive and systematic search was conducted across multiple electronic databases, including PubMed, Embase, and the Cochrane Library. This extensive search, covering publications from their inception to April 2023, ensured that a wide range of studies were considered, thus minimizing the risk of missing relevant research.

Inclusion and Exclusion Criteria: Clear and specific inclusion and exclusion criteria were established and adhered to throughout the selection process. These criteria were based on study type (observational studies), the association being studied (mild traumatic brain injury and Parkinsonism), and language (English). By setting these criteria, this study ensured that only studies relevant to the research question were considered, thus enhancing the focus and relevance of this meta-analysis.

Manual Search of Reference Lists: In addition to electronic database searches, manual searches of reference lists from relevant articles and review papers were conducted. This step helped identify additional studies that might not have been captured in the database searches, further reducing the risk of missing pertinent research.

Independent Review and Data Extraction: Two independent reviewers were involved in the data extraction process. Each eligible study’s data, such as the study design, sample size, definitions of mTBI and Parkinsonism, and outcomes, were independently extracted and cross-verified. Discrepancies were resolved through consensus or by consulting a third reviewer. This approach minimized subjective bias and ensured a thorough and unbiased evaluation of each study.

Quality Assessment of Included Studies: The quality of each included study was rigorously assessed using the established criteria. This assessment considered factors like study design, methodology, sample size, and potential biases. Figure 2a,b show graphical representations of the quality assessment, indicating that the studies included in this meta-analysis were of high quality and had a minimal risk of bias.

Exclusion of Duplicates and Irrelevant Studies: During the screening process, the duplicates and studies that did not meet the inclusion criteria were systematically excluded. This step ensured that only unique and relevant studies were considered for the final analysis.

Rigorous Full-Text Assessment: For studies that passed the initial screening, a full-text assessment was conducted to confirm their eligibility based on the predefined criteria. This thorough review process ensured that only studies that closely aligned with the research question and met the quality standards were included.

Overall, the selection process was designed to minimize the risk of bias and ensure that only relevant studies were included in the meta-analysis. The rigorous approach used in this study enhanced the validity and reliability of the findings, thus providing a solid basis for future research and clinical practice. There were no concerns about the quality of the included studies (Figure 2a,b).

### 3.1. Review: Narrative Summary

#### 3.1.1. Parkinsonism

Gardner et al. [5] found that military veterans with a history of mild traumatic brain injury (mTBI) had a 56% increased risk of developing Parkinson’s disease (PD) compared to those without a history of traumatic brain injury (TBI). Similarly, Phelps et al. [9] found that former college football players who experienced repetitive head impacts had a higher prevalence of neurological and neuropsychiatric conditions, including cognitive impairment, hypercholesterolemia, and Parkinsonism. In contrast, the association between mTBI and PD did not reach statistical significance in the study by Gardner et al. [10]. Nevertheless, the study did find an increased risk of PD after TBI compared to non-TBI trauma. Seidler et al. [6] conducted a case–control study and investigated the possible etiologic relevance to Parkinson’s disease (PD) of rural factors such as farming activity, pesticide exposure, well-water drinking, and animal contact; toxicologic exposure such as wood preservatives, heavy metals, and solvents; general anesthesia; head trauma; and differences in the intrauterine environment. The study found significantly elevated odds ratios for pesticide use, particularly for organochlorines and alkylated phosphates, but no association was present between PD and head injury. Taylor et al. [11] investigated the impact of environmental factors on PD risk and estimated the chronology, frequency, and duration of those exposures associated with PD; they also investigated the effects of family history on PD risk. The study found that head injury, a family history of PD, a history of tremors, and a history of depression were associated with an increased risk for PD. A mean latency of 36.5 years passed between the age of the first reported head injury and PD onset. Goldman et al. [12] conducted a case–control study in 93 twin pairs discordant for PD and found that a prior head injury with amnesia or loss of consciousness was associated with an increased risk of PD. The association was enhanced by truncating observations ten years before PD onset. The study suggested that a mild-to-moderate closed head injury may increase PD risk decades later. In a case–control study by Tsai et al. [13], head injury was found to be a significant risk factor for developing young-onset Parkinson’s disease. Similarly, McCann et al. [14] found that rural residency and a history of head injury were significant risk factors for PD. In contrast, Kuopio et al. [15] found no significant association between head injury and PD but identified other environmental risk factors, including domestic animal exposure, physical activity, and smoking. Baldereschi et al. [16] reported a significant association between pesticide exposure and PD. Smargiassi et al. [17] conducted a case–control study on 86 patients with Parkinson’s disease and 86 controls to identify the risk factors for the disease. The study identified that cranial trauma (OR: 2.88) was a potential risk factor for the development of Parkinson’s disease. Bower et al. [18] investigated the association between Parkinson’s disease and head trauma in a case–control study, identifying that head trauma was a significant risk factor for the development of Parkinson’s disease (OR = 4.3) and that the risk increased with the severity of the head trauma [18]. Dick et al. [19] conducted a case–control study in five European countries, revealing that exposure to pesticides (OR = 1.41) and having experienced multiple traumatic losses of consciousness (OR = 2.53) were significantly associated with Parkinson’s disease. Finally, Rugbjerg et al. [20] conducted a population-based case–control study that found a 50% increase in the prevalence of hospital contacts for a head injury before the first registration of Parkinson’s disease in the population (OR = 1.5), with the increased risk observed almost exclusively in injuries occurring in the three months preceding the first record of Parkinson’s disease [20]. A case–control study in Eastern India found that exposure to environmental risk factors, such as pesticides and toxins other than herbicides, increased the risk of Parkinson’s disease (PD). In contrast, head trauma was not a significant risk factor [21]. Similarly, a case–control study conducted in southern Italy did not find a significant association between head trauma and the risk of PD [22]. Another case–control study conducted in England did not find a significant association between early head injury and the risk of PD [23]. A nested case–control population-based analysis conducted in Minnesota did not find any association between TBI and subsequent α-synucleinopathies, including PD, dementia with Lewy bodies (DLBs), PD dementia (PDD), and multiple system atrophy (MSA) [24]. Furthermore, Nguyen et al. [25] found that a history of TBI was associated with an earlier onset of dementia with Lewy bodies. Some studies [26] suggest that mild traumatic brain injury, including cranial trauma, head trauma, and traumatic loss of consciousness, may be a potential risk factor for the development of Parkinson’s disease.

#### 3.1.2. Tic Disorder

Ranjan et al. [27] reported a case of an adult man who developed tics one year after a severe TBI as a rare complication of TBI. Individuals with early-onset post-traumatic tics may have had a previously unrecognized, mild tic disorder or a genetic predisposition to tics, which the TBI unmasked. In contrast, late post-traumatic tics could be due to the delayed effects of the injury on neural circuits connecting the frontal cortex and basal ganglia.

#### 3.1.3. Akathisia

Desai et al. [28] described a case of severe akathisia in a 13-year-old boy following a fall from a bicycle, which was most likely due to a single administration of haloperidol used to sedate the patient before a head CT scan. The literature on akathisia in pediatric patients, especially in patients following acute head injury, was reviewed, with suggestions for an approach to deal with these symptoms in this clinical setting.

#### 3.1.4. Meta-Analysis

A total of 18 studies were included in this meta-analysis. The odds ratio for Parkinson’s disease after mild traumatic brain injury was 1.717 (95% CI: 1.126–2.623), with a *p*-value of 0.01 and high heterogeneity (I^2^). The forest plot of the odds ratios for individual studies is shown in Figure 3.

Regarding other conditions such as progressive supranuclear palsy (PSP), multiple system atrophy (MSA), and corticobasal degeneration (CBD), the odds ratio was 1.637 (95% CI: 0.760–3.523), with z = 2.86 and *p* = 0.46, indicating no significant association between mild traumatic brain injury and the development of these conditions. The forest plot of the odds ratios for individual studies is shown in Figure 3.

The overall odds ratio for all conditions, including Parkinson’s disease and Parkinson plus syndromes, was 1.637 (95% CI: 1.065–2.515), with z = 2.86 and *p* = 0.01, indicating a significant association between mild traumatic brain injury and the development of these conditions. The forest plot of the odds ratios for individual studies is shown in Figure 3. Sensitivity analyses were performed by excluding individual studies one at a time, and the results did not change substantially.

Overall, this meta-analysis suggests that mild traumatic brain injury is associated with an increased risk of Parkinson’s disease, but there are only a few case reports on tics and akathisia after an mTBI. Further research is needed to confirm these findings and explore the potential mechanisms underlying the association between mild traumatic brain injury and the development of Parkinson’s disease.

#### 3.1.5. DLBs, Tics, and Akathisia

Lewy body dementia (LBD) is a type of dementia that can occur after a traumatic brain injury (TBI). It is characterized by Lewy bodies: the abnormal protein deposits that accumulate in the brain [4]. The study by Nguyen investigates the relationship between a history of traumatic brain injury (TBI) with loss of consciousness (LOC) and the age of onset of dementia with Lewy bodies (LBDs). Data from 576 subjects diagnosed with DLBs were obtained from the National Alzheimer’s Coordinating Center (NACC), and those with a history of remote TBI were compared to those without a history of TBI. The study found that those with a history of TBI had a slightly earlier clinician-estimated age of onset and age of diagnosis of DLBs compared to those without a history of TBI. Still, the differences did not reach statistical significance. The study suggests that remote TBI with LOC is not significantly associated with DLB onset. However, it is a significant risk factor for cognitive decline and earlier age of onset in other neurodegenerative conditions. The study suggests that further investigations into other potential risk factors for DLBs are warranted.

Tics are involuntary, repetitive movements or vocalizations often associated with Tourette Syndrome. However, tics can also occur after a TBI, particularly in children. A case report by Ranjan described an adult man who developed motor and vocal tics one year after a severe TBI. The cause of post-traumatic tics is not well understood, but they may be due to changes in brain function or structural damage to the basal ganglia, which are involved in the control of movement.

Akathisia is a movement disorder characterized by motor restlessness, the inability to sit still, and a feeling of inner tension. It can occur as a side effect of medications used to treat psychiatric disorders, such as antipsychotics. However, akathisia can also occur after a TBI. Desai reported the case of a 13-year-old boy who developed severe akathisia following a head CT scan that was preceded by haloperidol, a medication used to sedate the patient. The exact cause of post-traumatic akathisia is not well understood, but it is thought to be related to damage to the frontal cortex, which is involved in regulating movement.

In conclusion, while Parkinsonism has been studied, mild TBI, LBD, tics, and akathisia are less frequently reported in the literature. Nonetheless, these movement disorders can occur after a TBI and may significantly impact the quality of life of affected individuals. Further research is needed to better understand the underlying mechanisms of these conditions and to develop effective treatments.

#### 3.1.6. Publication Bias

Based on the visual inspection of the funnel plot (Figure 4), Egger’s *p*-value of 0.8107, and Begg’s test *p*-value of 0.4717, there was no evidence of publication bias in this meta-analysis. Egger’s test is a statistical method used to detect publication bias by evaluating the relationship between the effect size and standard error. On the other hand, Begg’s test is a non-parametric rank correlation test that assesses the correlation between the effect size and its variance. A non-significant *p*-value for both tests suggests that publication bias is unlikely to be a significant factor affecting the results of this meta-analysis. However, it is essential to note that these tests have low power, mainly when the number of studies included in the analysis is small, and other sources of bias, such as selective reporting, cannot be completely ruled out.

## 4. Discussion

The role of mild traumatic brain injury (mTBI) as a risk factor for Parkinson’s disease (PD) has been investigated in several studies. A study by De Michele et al. [22] in southern Italy found a positive association between a family history of PD or essential tremors and PD. TBI was not associated with subsequent α-synucleinopathies in general or any individual α-synucleinopathy in a population-based analysis conducted by Hasan et al. [24]. However, a case report by Nguyen described a patient who developed dementia with Lewy bodies one year after mTBI, suggesting that late post-traumatic tics could be due to delayed effects of the injury on the neural circuits connecting the frontal cortex and basal ganglia.

The present meta-analysis aimed to investigate whether mTBI is a risk factor for Parkinsonism. The analysis included 18 studies that were deemed appropriate for inclusion. The results suggest a significant association between mTBI and Parkinsonism, with an overall odds ratio of 1.637 (z = 2.86, *p* = 0.01) and an odds ratio of 1.717 (a = 3, *p* = 0.01) for PD specifically.

These findings align with previous studies examining the relationship between traumatic brain injury (TBI) and Parkinsonism, including a recent population-based analysis that found that TBI is associated with an increased risk of developing PD [24]. The disruption of brain function caused by TBI may lead to the accumulation of the alpha-synuclein protein, a hallmark of Parkinsonism [28].

While it is true that a history of mTBI has been associated with an increased risk of Parkinsonism, it is important to note that this correlation does not necessarily indicate a causative relationship. Instead, other factors may be contributing to the observed association. There are several possible explanations for the observed association between mTBI and Parkinsonism, such as common underlying risk factors, including a genetic predisposition and additional environmental exposures.

In light of the existing literature and the significant findings from our meta-analysis, which indicates an odds ratio of 1.637 (z = 2.86, *p* = 0.01) for the association between mild traumatic brain injury (mTBI) and Parkinsonism, the discussion inevitably extends to several other key domains. First, it is crucial to delve into the neurobiological mechanisms that could underpin this association. Post-mTBI pathophysiological changes such as neuroinflammation, oxidative stress, and blood–brain barrier disruption may serve as conduits to the neurodegenerative changes emblematic of Parkinsonism, including the accumulation of the alpha-synuclein protein. The dopaminergic system, which undergoes significant neuronal loss in Parkinson’s disease, merits particular attention as mTBI could act as an accelerant to the degenerative process in these neurons. Genetic predisposition also looms as a potential confounding factor; genes like SNCA and LRRK2, previously linked to Parkinson’s disease, could make certain individuals more susceptible to Parkinsonism following mTBI. Beyond biological considerations, the role of comorbidities such as diabetes and hypertension, alongside lifestyle elements like smoking and alcohol consumption, warrant further exploration to elucidate their potential synergistic impact on the risk of developing Parkinsonism post-mTBI. Additionally, psychological sequelae like depression and anxiety, common in both post-mTBI patients and individuals with Parkinson’s disease, could contribute to the risk profile via neurobiological stress pathways. Gender-based variations in susceptibility to Parkinsonism post-mTBI also represent an unexplored avenue that could have substantial implications for preventive strategies. From a clinical standpoint, these findings necessitate the development of early screening procedures, targeted prevention measures, and perhaps even pharmacological interventions aimed at pathways identified in future research. On a broader scale, these findings could inform public health policies, potentially instigating more stringent regulations around activities with high risks of head injury, such as certain contact sports and occupational settings. In conclusion, while our study underscores a significant association between mTBI and Parkinsonism, it also necessitates a multi-dimensional discourse encompassing biological, psychological, and socio-environmental factors to fully understand this complex relationship, thereby laying the groundwork for future multi-faceted research and intervention strategies.

In conclusion, while an association between a history of mTBI and an increased risk of Parkinsonism has been observed, it is important to consider other possible explanations for this relationship, including shared risk factors, confounding factors, and bias. Additional research and experimentation are needed to determine whether a causal relationship exists between mTBI and Parkinsonism.

## 5. Limitations

Despite the significant findings of this meta-analysis, several limitations must be considered.

First, the studies included in this analysis used different definitions and criteria for mild traumatic brain injury (MTBI) and Parkinsonism, which may have affected the consistency of the results. The studies varied in their sample sizes, follow-up periods, and population demographics. This heterogeneity in methodology and patient characteristics can introduce variability into the findings, potentially limiting the generalizability of the results. The variation in the definitions of mTBI, ranging from concussion to more severe forms of head injury, and the differing diagnostic criteria for Parkinsonism across studies may have led to a broad spectrum of patient profiles, influencing the strength and direction of the observed associations.

Second, although efforts were made to identify and include all relevant studies, some may be as yet unnoticed, potentially leading to publication bias. However, Egger’s and Begg’s tests did not indicate significant publication bias in this meta-analysis. Nonetheless, it is important to acknowledge that these tests have limited power, especially in meta-analyses with a small number of studies. Additionally, the inclusion of only English language studies could introduce a language bias, potentially excluding relevant non-English research and limiting the comprehensiveness of the analysis.

Third, the studies included in this analysis were primarily observational, which cannot establish causality. Additionally, some studies relied on self-reported data, which may be prone to recall bias. This is particularly relevant in retrospective studies where participants may inaccurately recall past events or injuries, leading to the misclassification of exposure status. The observational nature of the studies also meant that they were subject to confounding factors that could influence the observed associations.

Fourth, the studies varied in assessing potential confounding factors, such as age, sex, and comorbidities, which may have influenced the results. The lack of uniformity in controlling for these confounders across studies can lead to residual confounding, thereby affecting the reliability of the findings. For instance, age and sex are known to influence the risk of both mTBI and Parkinsonism, and their inconsistent consideration across studies could skew the results.

Finally, this meta-analysis focused specifically on the association between mTBI and Parkinsonism and did not consider the other potential risk factors or the potential interplay between different risk factors. This narrow focus may overlook the multifactorial nature of Parkinsonism, where genetic, environmental, and lifestyle factors, along with mTBI, can collectively contribute to the disease’s development [29]. The interaction between mTBI and these other risk factors remains an area for further exploration.

In conclusion, while the findings of this meta-analysis suggest that mTBI may be a significant risk factor for Parkinsonism, further research is needed to confirm these results and to better understand the mechanisms underlying this relationship. Future studies should aim for more standardized definitions and methodologies, consider a broader range of risk factors, and employ designs that are capable of establishing causal relationships.

## Figures and Tables

**Figure 1 life-14-00032-f001:**
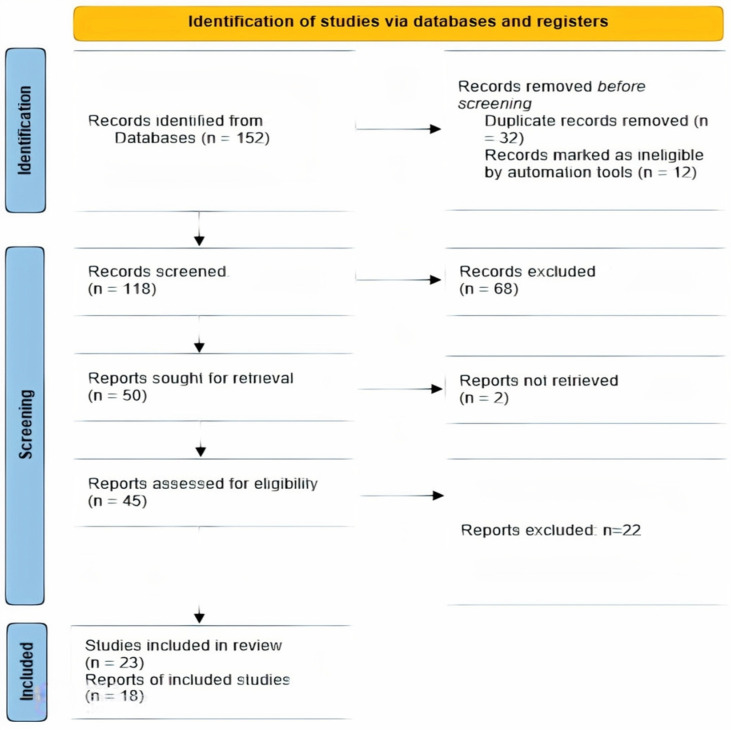
PRISMA flow diagram showing the selection process of the studies included in the present analysis.

**Figure 2 life-14-00032-f002:**
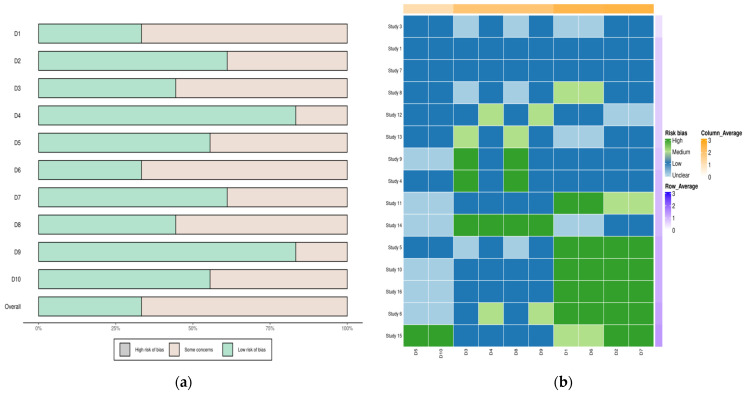
(**a**) Risk of bias analysis heatmap for each study; (**b**) Summary of risk of bias.

**Figure 3 life-14-00032-f003:**
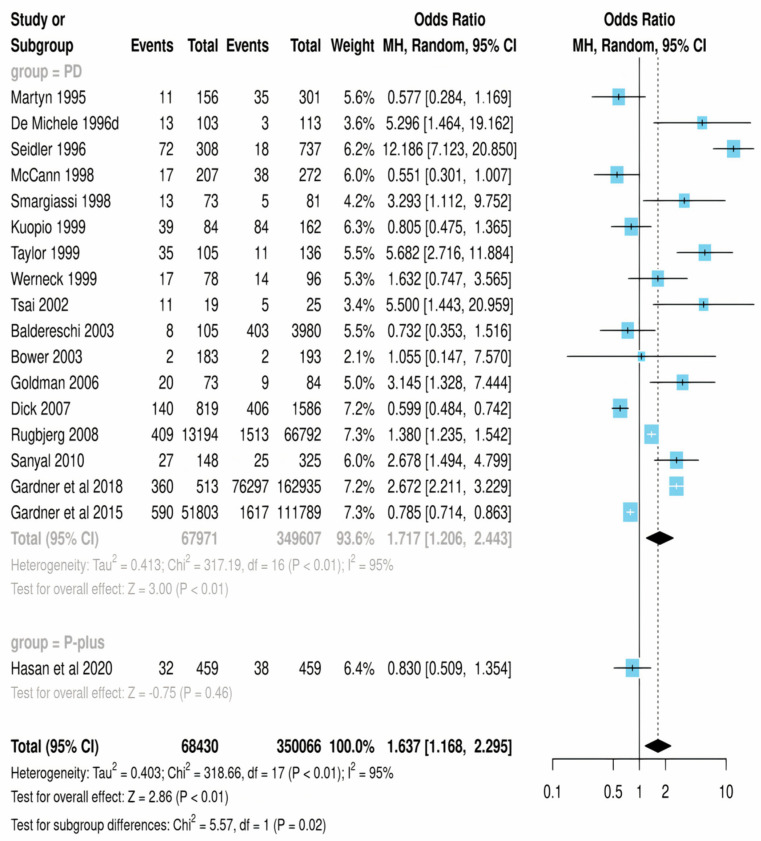
Forest plot of the prevalence of PD and PD plus syndromes in patients who suffered a TBI.

**Figure 4 life-14-00032-f004:**
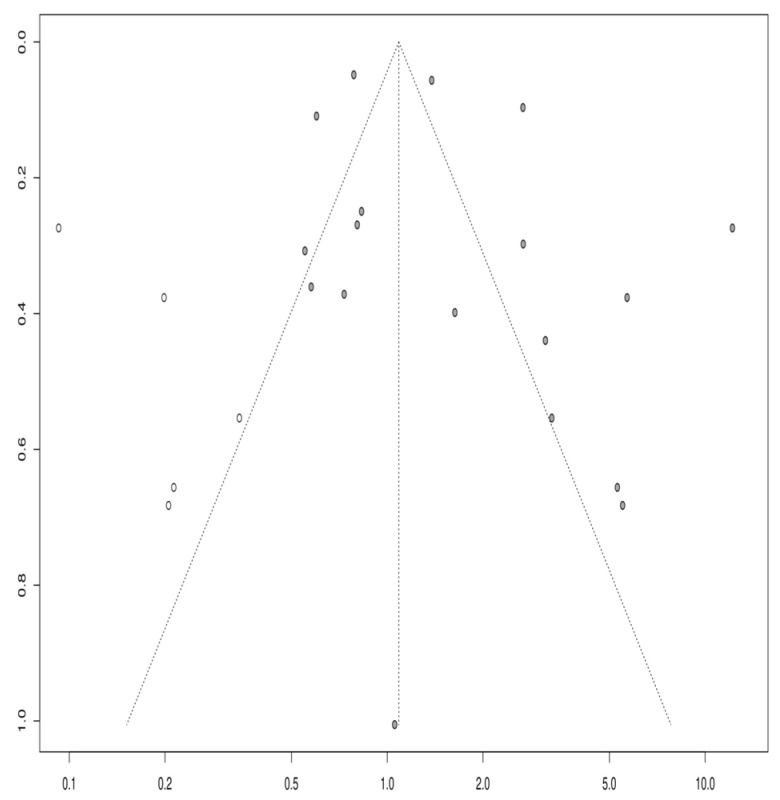
Funnel plot showing a low risk of publication bias. White circles are indicating the observed studies.

## Data Availability

Data are available upon request from the correspondence author.

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
