# Peer review of "Mild Traumatic Brain Injury as a Risk Factor for Parkinsonism, Tics, and Akathisia: A Systematic Review and Meta-Analysis"

_life, 2023, doi:10.3390/life14010032_

Round 1

Reviewer 1 Report

Comments and Suggestions for Authors

In this paper, Nashaba et al. have reported that Mild Traumatic Brain Injury as a Risk Factor for Parkinsonism, tics and Akathisia. It is interesting, but there are some shortcomings.

1.     The location of the occurrence of mild traumatic brain injury is not reflected in this paper. The main affected area of Parkinson's disease is the midbrain region. Therefore, it is necessary to clarify the specific brain area where mild traumatic brain injury occurs, and then conduct correlation analysis with the incidence of Parkinson's disease.

2.     Mild traumatic brain injury may be related to many neurodegenerative diseases, and the author needs to conduct a comprehensive analysis of more disease types in order to obtain results that better support the conclusions of this study.

3.     There are some spelling errors in the article, please make corrections.

Comments on the Quality of English Language

There are some spelling errors in the article, please make corrections.

Author Response

Enclosed please find the response.

Reviewer 2 Report

Comments and Suggestions for Authors

The article carries out a meta-analysis aimed to assess the association between mild traumatic brain injury (mTBI) and the risk of developing Parkinsonism. It is an article that is well crafted.

In the introduction, we recommend citing this article: DOI: 10.3233/JPD-202067

Regarding its development, it follows a methodology with PRISMA criteria, which is appropriate, and so are the results presented.

On the other hand, it would be interesting to collect what type of mTBI, that is, in what part of the brain, or point out that it is not generalized, since I understand that frontal damage is not the same as occipital, temporal, or parietal damage.

We would like the conclusions to be after the limitations, and to be as humble as possible, since this study, although interesting, is very risky, as it relates traumatic brain damage to parkinsonism.

Author Response

Enclosed please find the response.

Reviewer 3 Report

Comments and Suggestions for Authors

1.      Some studies relied on self-reported data, which may be prone to recall bias. This can affect the accuracy of the information regarding the history of mTBI and subsequent development of Parkinsonism.

2.      The studies included in the meta-analysis used different definitions and criteria for mild traumatic brain injury (mTBI) and Parkinsonism. This variability could affect the consistency and comparability of the results.

3.      There is mention of heterogeneity among the studies included in the meta-analysis. High heterogeneity can reduce the reliability of the meta-analysis results

4.      While the article discusses the association between mTBI and Parkinsonism, it does not delve deeply into the underlying mechanisms of this relationship. Understanding these mechanisms is crucial for developing targeted interventions.

5.      Clarify that the observed association between mTBI and Parkinsonism does not necessarily imply a causal relationship. Emphasize the need for prospective studies or experimental designs to establish causation.

6.      Limiting the inclusion of studies to those published exclusively in English may introduce language bias. Given the global scope of research, excluding non-English studies may curtail the incorporation of diverse perspectives and findings, potentially compromising the study’s inclusivity.

7.      In the methods, details about the extraction process are missing.

8.      The meta-analysis acknowledges significant overall heterogeneity but falls short in providing a thorough examination of heterogeneity within distinct subgroups, including age groups or levels of injury severity.

9.      Although the discussion mentions the need for further exploration of gender-based variations, it fails to provide any meaningful insights or hypotheses on how gender might influence the observed association.

10.  Provide a brief acknowledgment of how methodological variations in the studies (sample sizes, follow-up periods) could contribute to the heterogeneity observed in the meta-analysis.

11.  It would be preferable for the manuscript to include information about Akathisia in the introduction section.

12.  In the methods section of the manuscript, the I-squared statistic is mentioned. It would be beneficial to briefly explain what the I-squared statistic is and discuss the rationale behind selecting it.

13.  In the results section, provide additional details on the criteria and the selection process for studies included in the meta-analysis, particularly emphasizing aspects that enhance their relevance.

14.  In Figure 2, particularly in section A, and Figure 3, enhance reader comprehension by providing additional explanations and summarizing the key points of each figure.

15.  The inclusion of 18 studies in this meta-analysis represents a relatively small proportion compared to the total number of articles available.

16.  In line 239 and line 240, the manuscript refers to the Egger’s p-value and Begg’s test p-value, respectively. Provide a more detailed clarification on which of these values holds greater reliability and the rationale for this choice, considering the disparity between the two p-values, despite both being non-significant. Additionally, specify which of these tests was utilized for Figure 3.

17.  Various articles employ different methods to demonstrate significance. The manuscript should clarify that these differences in presentation methods do not impact the results of the current study.

18.  Provide titles for X and Y axis in figure 4.

19.  The results mentioned from Baldereshci’s article aren’t relevant line 139-140 as they do not mention anything about head injuries or mTBI. The same article does provide more relevant information for your manuscript such as “Head injury was inversely associated with time from diagnosis to mortality among PD patients who participated in sports (participation: HR, 0.53; 95% CI, 0.28–1.04; no participation: HR, 0.93; 95% CI, 0.46–1.91), but head injury was not associated with mortality among controls.”

20.  Paragraph 200-203 mentions that TBI isn’t associated with tics and akathisia yet the results provided in sections “akathisia”, “tic disorder” and “DLB, Tics, and akathisia” contradict that.

21.  Line 204 typo, DLB is written to refer to Lewy Body Dementia instead of LBD

Author Response

Enclosed please find the response.

Round 2

Reviewer 1 Report

Comments and Suggestions for Authors

The manuscript did not show significant improvement.

Reviewer 3 Report

Comments and Suggestions for Authors

Thank you for taking all comments into consideration.